# Inefficient Induction of Neutralizing Antibodies against SARS-CoV-2 Variants in Patients with Inflammatory Bowel Disease on Anti-Tumor Necrosis Factor Therapy after Receiving a Third mRNA Vaccine Dose

**DOI:** 10.3390/vaccines10081301

**Published:** 2022-08-11

**Authors:** Paola López-Marte, Alondra Soto-González, Lizzie Ramos-Tollinchi, Stephan Torres-Jorge, Mariana Ferre, Esteban Rodríguez-Martinó, Esther A. Torres, Carlos A. Sariol

**Affiliations:** 1Gastroenterology Research, School of Medicine, University of Puerto Rico, San Juan 00925, Puerto Rico; 2Department of Microbiology and Medical Zoology, School of Medicine, University of Puerto Rico, San Juan 00925, Puerto Rico; 3Division of Gastroenterology, School of Medicine, University of Puerto Rico, San Juan 00925, Puerto Rico

**Keywords:** COVID-19 variants, COVID-19 vaccine, IBD, ulcerative colitis, Crohn’s disease, anti-TNF

## Abstract

Management of inflammatory bowel disease (IBD) often relies on biological and immunomodulatory agents for remission through immunosuppression, raising concerns regarding the SARS-CoV-2 vaccine’s effectiveness. The emergent variants have hindered the vaccine neutralization capacity, and whether the third vaccine dose can neutralize SARS-CoV-2 variants in this population remains unknown. This study aims to evaluate the humoral response of SARS-CoV-2 variants in patients with IBD 60 days after the third vaccine dose [BNT162b2 (Pfizer-BioNTech) or mRNA-1273 (Moderna)]. Fifty-six subjects with IBD and 12 healthy subjects were recruited. Ninety percent of patients with IBD (49/56) received biologics and/or immunomodulatory therapy. Twenty-four subjects with IBD did not develop effective neutralizing capability against the Omicron variant. Seventy percent (17/24) of those subjects received anti-tumor necrosis factor therapy [10 = adalimumab, 7 = infliximab], two of which had a history of COVID-19 infection, and one subject did not develop immune neutralization against three other variants: Gamma, Epsilon, and Kappa. All subjects in the control group developed detectable antibodies and effective neutralization against all seven SARS-CoV-2 variants. Our study shows that patients with IBD might not be protected against SARS-CoV-2 variants, and more extensive studies are needed to evaluate optimal immunity.

## 1. Introduction

As the coronavirus pandemic is evolving, more questions and challenges have arisen. COVID-19 disease is caused by severe acute respiratory syndrome coronavirus 2 (SARS-CoV-2), with a 2–3% fatality rate [1]. Vaccination against this virus has successfully reduced symptomatic COVID-19 infection, hospitalizations, and mortality [2]. Yet, the characterization of the vaccine’s efficacy among immunocompromised patients is still unclear. Patients with inflammatory bowel disease (IBD) are part of this immunocompromised population. As is well known, initial clinical trials for the vaccines against COVID-19 excluded immunosuppressed patients [3]. Thus, whether these vaccines can induce an adequate and long-lasting immune response against the virus in this population is not fully addressed. Inflammatory bowel diseases, including Crohn’s disease (CD) and ulcerative colitis (UC), are complex, chronic, and costly conditions characterized by an immune-mediated inflammatory response in the gastrointestinal (GI) tract. Although the etiology is unknown, it is thought to be multifactorial, including environmental triggers, gut microbiota, and immune dysregulation in genetically susceptible individuals [4]. Management of these diseases often relies on immunomodulatory and biological agents to achieve and maintain clinical, biochemical, and endoscopic remission in these patients, placing these individuals vulnerable to bacterial and viral infections [5]. These drugs also impair the protective immune response elicited by various vaccines [3]. For instance, it has been shown that infliximab reduces immunity to hepatitis B, hepatitis A, pneumococcal, and influenza vaccinations [6].

The emergence of SARS-CoV-2 variants has posed a significant concern in the general community, especially in patients with dysregulated immune responses and comorbidities. Mutations modifying the spike protein structure can consequently alter the protein and the human ACE-2 receptor interaction, further changing immune response and threatening the efficacy of mRNA-based COVID-19 vaccine against variants [7]. The administration of the third dose of mRNA COVID-19 vaccine has been promoted to boost the immune response against SARS-CoV-2 rapidly appearing variants [8].

Recent evidence showing the efficacy of COVID-19 immunization in patients with IBD has suggested that they can reach seroconversion against the virus after two doses of mRNA-COVID-19 vaccine [9]. However, with the emergent SARS-CoV-2 variants, whether a third vaccine dose can provide the same efficacy remains unknown. In this study, we report the results of the humoral immune response against seven SARS-CoV-2 variants of concern in patients with IBD on biological and/or immunomodulatory therapies 60-days after receiving three doses of mRNA-COVID-19 vaccine.

## 2. Materials and Methods

Patients ≥ 21 years of age diagnosed with CD or UC exposed and unexposed to biologic and/or immunomodulatory therapy were recruited between October 2021 and May 2022 at the University of Puerto Rico IBD Clinics. Patients with Hermansky-Pudlak syndrome and pregnant women were excluded from the study. Blood samples were collected at 60 ± 7 days after receiving the third dose of mRNA COVID-19 vaccine [BNT162b2 (Pfizer-BioNTech) or mRNA-1273 (Moderna)]. A control group of healthy adult volunteers matched for vaccination status was obtained from subjects participating in the clinical protocol “Molecular Basis and Epidemiology of Viral Infections circulating in Puerto Rico”.

### 2.1. Anti-Spike IgG Levels by ELISA

Anti-Spike IgG levels were measured with an indirect in-house ELISA for the semi-quantitative determination of human IgG antibody class [10,11]. Briefly, 96-well microplates were coated overnight at 4 °C with 2 μg/mL of recombinant SARS-CoV-2 S1-RBD (GenScript, Piscataway Township, NJ, USA) protein in a carbonate-bicarbonate buffer. After washing and blocking, samples (serum or plasma) were diluted at 1:100 and incubated at 37 °C for 30 min. Following a washing step, horseradish peroxidase (HRP) labeled-mouse anti-human IgG-Fc (GenScript, Piscataway Township, NJ, USA) was added and incubated for 30 min at 37 °C. After washing, the substrate solution was added; the reaction was stopped with 10% HCl, and the absorbance was measured at 492 nm (A492) using a Multiskan FC reader (Thermo Fisher Scientific). Samples with A492 > 0.499 were considered positive.

### 2.2. Neutralizing Antibodies against SARS-CoV-2 Variants of Concern

For the neutralizing activity, the cPassTM SARS-CoV-2 neutralization antibody detection kit (GenScript, Piscataway Township, NJ, USA) was used to measure inhibitory capability based on the ability of the antibodies to target the host ACE2 receptor and viral receptor-binding domain (RBD) interaction. Serum or plasma samples were diluted according to the manufacturer’s instructions and incubated with a soluble SARS-CoV2 receptor binding domain (RBD-HRP) antigen for 30 min, mimicking a neutralization reaction. Following incubation, samples were added to a 96-well plate coated with human ACE-2 protein. Since this is an inhibition assay, color intensity is inversely proportional to the amount of neutralizing antibodies present in samples [10,12].

This test was performed on SARS-CoV-2 Wild type and seven SARS-CoV-2 variants of concern: Alpha, Beta, Gamma, Epsilon, Kappa, Delta, and Omicron. A result of ≥30% in a surrogate virus neutralization test (sVNT%) demonstrated an effective viral variant neutralization capacity. Results were stratified by the mechanism of action of the specific immunosuppressive therapy and compared to a healthy control group. Informed consent was obtained from all subjects involved in the study.

### 2.3. Statistical Analysis

Analyses were performed using *Intellectus Statistics* online software (Clearwater, FL, USA). Categorical data items were summarized using frequency counts and percentages, while continuous quantitative variables were described as mean ± standard deviation (SD). The comparative two-tailed Mann–Whitney two-sample was performed among the two cohorts, patients with IBD and the control group, to evaluate statistical differences in the mean values for anti-Spike IgG Levels and sVNT% of the SARS-CoV-2 variants of concern. There were 56 observations in participants with IBD and 12 observations in controls. In addition, we evaluated differences in IgG levels and sVNT% among patients with IBD stratified by their current medication by conducting a Kruskal–Wallis test. Post-hoc analyses were completed for statistically significant differences with Pairwise comparisons. Statistical significance for all analyses was set at alpha < 0.05.

### 2.4. Ethical Statement

This study was approved by the University of Puerto Rico Medical Sciences Campus IRB (protocol: #1250121). Volunteers in the control group participated in the Advarra IRB-approved clinical protocol “Molecular Basis and Epidemiology of Viral Infections Circulating in Puerto Rico” (Pro0004333).

## 3. Results

Fifty-six subjects with IBD and 12 healthy controls are reported. Fifty-one of the 56 with IBD received biological and/or immunomodulatory drugs. Blood samples were examined 60 ± 7 days after the third vaccine dose in the IBD subjects and 60 ± 10 days after the third dose in the controls. 82% (46/56) of subjects with IBD had a diagnosis of CD, 61% were males (34/56), and 39% (22/56) were females. The mean age for the subjects with IBD was 42 ± 13.2 (41 ± 13.1 for males and 44 ± 13.5 for females). Most subjects received anti-TNF therapy (30/56), and one was on concomitant oral corticosteroids (see Table 1). All subjects with IBD developed detectable antibodies after 60 ± 7 days. Twenty-four (24) subjects with IBD did not develop neutralizing capability against the Omicron variant. Seventy-one percent (17/24) of those subjects were receiving anti-TNF therapy [10: adalimumab, 7: infliximab], two of them had a history of COVID-19 infection, and one subject did not develop neutralizing antibodies against three other variants: Gamma, Epsilon, and Kappa. In the healthy controls, only one subject had a prior history of COVID-19 infection. All developed detectable antibodies and effective humoral responses against all seven variants of SARS-CoV-2, although levels against the Omicron variant were lower than those against the other variants. However, all subjects in the control group developed more than 30% neutralization, which did not occur in the IBD group (see Figure 1a).

When analyzed by non-parametric tests, participants with IBD showed statistically lower values of sVNT% for the Gamma, Epsilon, Kappa, Delta, and Omicron variants when compared to healthy controls (*p* = 0.015, <0.001, <0.001, 0.014, 0.031, respectively), as detailed in Figure 1a. Data of anti-Spike IgG levels and sVNT% between the control group and patients with IBD was stratified by biological/immunomodulatory treatment, as seen in Figure 1b. We found statistical differences in the sVNT% values for SARS-CoV-2 Wild type (*p* = 0.008), and Beta (*p* = 0.029), Gamma (*p* = 0.001), Epsilon (*p* < 0.001), Kappa (*p* < 0.001), Delta (*p* = 0.018), and Omicron (*p* = 0.007), variants, mainly between controls and patients receiving anti-TNF therapy, detailed in Table 2 and Figure 1b.

## 4. Discussion

There are scarce data about the efficacy of a third vaccine dose of COVID-19 against SARS-CoV-2 variants in patients with IBD. Our results indicate that humoral immune response after the third vaccine dose in patients with IBD on anti-TNF therapy might not be protective against SARS-CoV-2 variants, particularly against Omicron. We show that participants with IBD had statistically lower values of sVNT% for the Gamma, Epsilon, Kappa, Delta, and Omicron variants compared to healthy controls. Differences are evident in the sVNT% values for SARS-CoV-2 Wild type, along with Beta, Gamma, Epsilon, Kappa, Delta, and Omicron variants, mainly between controls and patients receiving anti-TNF therapy.

Our findings after three doses of the vaccine against the Omicron variant are comparable with organ transplant recipients receiving immunosuppressive therapy [13] and other groups with autoimmune inflammatory disease receiving anti-TNF therapy after two vaccine doses [14]. The mechanism of how anti-TNF decreases immune response from vaccines is not fully understood. Yet, it is known that TNF influences important cellular behavior such as migration and proliferation. Thus anti-TNF drugs may interfere with germinal centers to induce an adequate humoral immune response [15].

The small sample limits these results in both cohorts and the short follow-up period included in the analysis. Among other limitations in our analysis were differences in gender and age between the participants with IBD and controls. In the control group, the female gender was predominant (*p* = 0.030) and the subjects were also younger [control: 24 years vs. IBD: 36 years (*p* = 0.050)]. Nevertheless, this would not explain our findings. It is important to note that the number of patients with IBD receiving anti-integrin therapy in our cohort is too small to reach statistical power and detect differences between groups.

Our data suggest that patients with IBD (especially those on biological medications) might benefit from an additional vaccine dose to produce vaccine-induced antibodies with a stronger and more effective neutralizing capability. Patients receiving anti-TNF may have less vaccine protection than those treated with other biologics, as seen in our findings. This report presents a snapshot of the rapid evolution of the SARS-CoV-2 variants that keep changing our current understanding of the viral behavior in response to the vaccine. More prospective studies with a larger sample and extended time frames are needed to ensure the optimal immunity for these high-risk patients.

## Figures and Tables

**Figure 1 vaccines-10-01301-f001:**
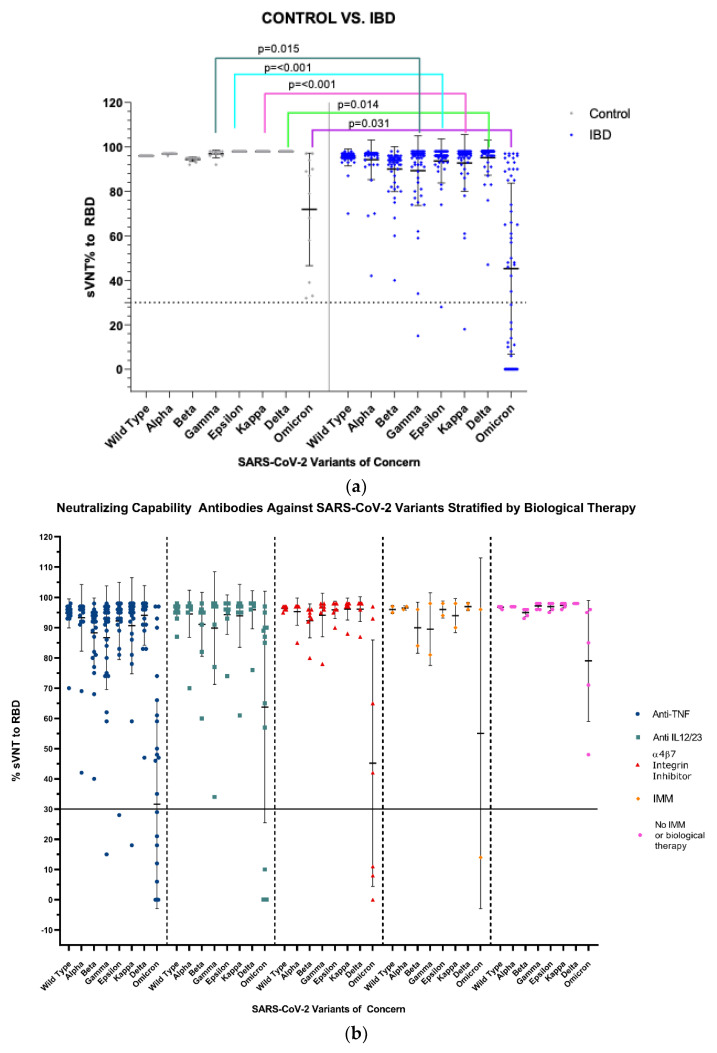
(**a**) Antibody neutralization against receptor binding domain (RBD) of SARS-CoV-2 variants of concern in controls vs. patients with IBD after 60-days of mRNA COVID-19 vaccine. (**b**) A drug therapy mechanism stratifies antibody neutralization against SARS-CoV-2 variants of concern in patients with IBD. All the healthy controls developed detectable antibodies and effective humoral responses against all seven variants of SARS-CoV-2. Although levels against the Omicron variant were lower than those against the other variants, they developed more than 30% neutralization, which did not occur in the IBD group. The dashed line represents the limit for effective antibody neutralization (≥30%), and the bars represent the mean with standard deviation. (**b**) Twenty-four (24) subjects with IBD did not develop effective neutralizing capability against the Omicron variant. Three subjects (3/24) were receiving anti-IL 12/23 inhibitor, the other three (3/24) subjects were receiving integrin inhibitor, and one (1/24) subject was on IMM. Seventy-one percent (17/24) of those subjects were receiving anti-TNF therapy [10 = adalimumab, 7 = infliximab], two of them (2/17) had a history of COVID-19 infection, and one (1/17) subject did not develop immune neutralization against three other variants: Gamma, Epsilon, and Kappa. The solid line represents the limit for effective antibody neutralization (≥30%), and bars represent the mean with standard deviation. Anti-TNF: anti-tumor necrosis factor, Anti-IL 12/23: anti-interleukin 12/23, a4b7 integrin inhibitor: alpha 4 and beta 7 integrin inhibitor, IMM: immunomodulator.

**Table 1 vaccines-10-01301-t001:** Characteristics of the subjects in the study.

	IBD	Control
Subjects, n	56	12
Mean age in years, n	42 ± 13.3	35 ± 16.9
Male, n (%)	34 (61%)	3 (25%)
Female, n (%)	22 (29%)	9 (75%)
Median BMI, median (IQR)	25.3 (22.8–29.2)	24.1 (21.5–32.7)
IBD Type		
Ulcerative colitis, n (%)	10 (18%)	-
Crohn’s disease, n (%)	46 (82%)	-
Prednisone ≥ 20 mg use daily, n	1	-
Biological Therapy use, n (%)	49 (87%)	-
Adalimumab, n	16	-
Infliximab, n	14	-
Anti-IL12/23, n	12	-
a4b7 integrin inhibitor, n	7	-
Immunomodulator use, n (%)	2 (4%)	-
Combination immunomodulator and biological use, n (%)	0	-
Salicylates, n (%)	2 (4%)	-
No therapy, n (%)	3 (5%)	-
History of COVID-19 infection, n (%)	6 (11%)	1 (8%)

**Table 2 vaccines-10-01301-t002:** Kruskal–Wallis Rank Sum Test for anti-Spike IgG levels and sVNT% stratified by therapy vs. control, and Pairwise Comparisons for the Mean Ranks.

Variable Levels	Mean	SD	n	*p*-Value
Anti-Spike IgG Levels				
Anti-TNF	2789.98	1663.38	30	0.055
IL12/23 Inhibitors	3957.58	1686.53	12	
Integrin α_4_β_7_ Inhibitor	3962.14	1664.47	7	
Non-Biologic	4448.00	1167.81	7	
Controls	4235.67	941.32	12	
Wild Type sVNT%				
Anti-TNF	94.67	4.78	30	0.008
IL12/23 Inhibitors	95.31	2.95	12	
Integrin α_4_β_7_ Inhibitor	96.40	0.73	7	
Non-Biologic	96.49	0.74	7	
Controls	96.18	0.18	12	
However, results indicated that none of the individual pairwise comparisons were significantly different.				
Alpha sVNT%				
Anti-TNF	93.32	11.11	30	0.072
IL12/23 Inhibitors	94.81	7.81	12	
Integrin α_4_β_7_ Inhibitor	95.51	4.45	7	
Non-Biologic	97.01	0.55	7	
Controls	96.93	0.19	12	
Beta sVNT%				
Anti-TNF	88.32	11.61	30	0.029
IL12/23 Inhibitors	91.03	10.54	12	
Integrin α_4_β_7_ Inhibitor	92.66	5.74	7	
Non-Biologic	93.76	4.44	7	
Controls	94.23	1.0	12	
However, results indicated that none of the individual pairwise comparisons were significantly different.				
Gamma sVNT%				
Anti-TNF	86.68	17.20	30	0.001
IL12/23 Inhibitors	89.76	18.74	12	
Integrin α_4_β_7_ Inhibitor	93.99	7.38	7	
Non-Biologic	95.16	6.28	7	
Controls	96.80	1.69	12	
Pairwise Comparison Anti-TNF vs. Non-Biologic	Obs. Diff. 23.74	Critical Diff. 23.30		
Anti-TNF vs. Controls	23.53	18.96		
The results of the multiple comparisons indicated significant differences between the following variable pairs: Anti TNF-Non-Biologic and Anti TNF-Controls				
Epsilon sVNT%				
Anti-TNF	92.13	12.64	30	<0.001
IL12/23 Inhibitors	94.17	6.61	12	
Integrin α_4_β_7_ Inhibitor	95.87	2.66	7	
Non-Biologic	96.70	1.45	7	
Controls	97.95	0.07	12	
Pairwise Comparison Anti-TNF vs. Controls	Obs. Diff. 33.68	Critical Diff. 18.96		
IL12/23 Inhibitors vs. Controls	29.46	22.66		
The results of the multiple comparisons indicated significant differences between the following variable pairs: Anti TNF-Controls and IL12/23 Inhibitors-Controls				
Kappa sVNT%				
Anti-TNF	90.73	15.92	30	<0.001
IL12/23 Inhibitors	93.87	10.48	12	
Integrin α_4_β_7_ Inhibitor	96.10	3.55	7	
Non-Biologic	96.34	2.77	7	
Controls	97.93	0.15	12	
Pairwise Comparison Anti-TNF vs. Controls	Obs. Diff. 35.60	Critical Diff. 18.96		
IL12/23 Inhibitors vs. Controls	26.92	22.66		
The results of the multiple comparisons indicated significant differences between the following variable pairs: Anti TNF-Controls and IL12/23 Inhibitors-Controls				
Delta sVNT%				
Anti-TNF	94.16	9.81	30	0.018
IL12/23 Inhibitors	96.11	6.43	12	
Integrin α_4_β_7_ Inhibitor	96.23	4.31	7	
Non-Biologic	97.68	0.84	7	
Controls	98.18	0.11	12	
Pairwise Comparison Anti-TNF vs. Controls	Obs. Diff. 19.87	Critical Diff. 18.96		
The results of the multiple comparisons indicated significant differences between Anti-TNF-Controls				
Omicron sVNT%				
Anti-TNF	31.60	34.57	30	0.007
IL12/23 Inhibitors	63.64	38.24	12	
Integrin α_4_β_7_ Inhibitor	45.10	40.67	7	
Non-Biologic	72.03	30.95	7	
Controls	72.00	25.38	12	
Pairwise Comparison Anti-TNF vs. Controls	Obs. Diff. 20.38	Critical Diff. 18.96		
The results of the multiple comparisons indicated significant differences between Anti-TNF-Controls				

## Data Availability

The authors store data supporting the reported results and are available upon request.

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
