# Peer review of "Inefficient Induction of Neutralizing Antibodies against SARS-CoV-2 Variants in Patients with Inflammatory Bowel Disease on Anti-Tumor Necrosis Factor Therapy after Receiving a Third mRNA Vaccine Dose"

_vaccines, 2022, doi:10.3390/vaccines10081301_

Round 1
Reviewer 1 Report
This study attempts to understand the humoral response following COVID-19 vaccine booster in patients with IBD, with or without concomitant use of immunomodulators such as TNF-alpha inhibitors. Authors enrolled 56 IBD patients and had 12 healthy controls. Compared to controls, IBD patients developed lower neutralizing antibodies titers against Omicron, and majority of which were also on immunomodulators.
Although relatively small sample size and using surrogate endpoints, this is a significant proof of concept study that may help us better understand immune response to COVID-19 amongst IBD patients. Nevertheless, findings may need to be validated in a larger study.
This study evaluated neutralizing titers against wild type, Alpha, Beta, Gamma, Kappa, Epsilon, Delta, and Omicron variants. Other than titers vs Omicron, these are nice to know but not as relevant now as the dominant variants are Omicron subvariants. Would the authors be able to add neutralization titers against BA.4, BA.5, and BA.2.12.1 (the dominant subvariants to date)?
Figure 1 appears to show neutralizing antibody titers against Omicron variant is diminished in both IBD and health controls compared to response to other variants. Maybe this should be commented in the results?
Figure 1 both (a) and (b) – Need to add labels. What does the bar symbolize? Mean, geometric mean, or median? Did you use 95% CI or standard deviation or IQR? Also, add label in the graph to show which ones are statistically significant.
Figure 1 (b) – Right most set uses black dots similar to the line for the mean, which makes it hard to read. Maybe use a different color (not black) for the dots. Also, maybe it would be much more useful to have the bar with either 95% CI or SD or IQR?
Based on this study, does the author recommend a 2nd booster dose (e.g., fourth dose) for all IBD patients or for just a subset of IBD patients?
OTHER COMMENTS:
Line 19: “56” should be spelled out as its in the beginning of a sentence. Should read “Fifty six subjects…”
Line 20: Same issue as above. “90%” should be spelled out as its in the beginning of a sentence.
Line 20: Could be abbreviated further to: “Ninety percent of IBD patients (49/56)…”
Line 21: Same issue as above. “24” should be spelled out as it is in the beginning of a sentence.
Line 22: Same issue as above. Spell out “70%”
Line 22: anti-tumor necrosis factor should not be capitalized.
Line 22 and 23: Would remove the “=” sign.
Line 38-39: Not sure if I would agree fully with this statement. Initial registry clinical trials do exclude immunocompromised host, but subsequent studies albeit smaller then would target immunocompromised population.
Line 61: Did you mean: “mRNA based COVID-19 vaccine”?
Line 115: change “56” to “Fifty six” (spell out if start of a sentence).
Line 118: same issue as above, spell out “82%”.
Line 122: What does “effective neutralizing capability” mean? Subjects were not exposed to COVID-19 in this study to show that it was effective.
Line 123-124: remove “=” signs.
Line 125: “immune neutralization”? Maybe use “neutralizing antibodies”?
Reviewer 2 Report
The authors describe a case-control study for the efficiency of 3 doses of mRNA vaccines against SARS-CoV-2 in IBD patients. Although anti-Spike IgG Levels were not statistically different between healthy controls and IBD patients, the surrogate virus neutralization tests indicated the inefficient induction of neutralizing antibodies against some of the SARS-CoV-2 variants in IBD patients, especially treated with anti-TNF biologics. The results would be informative to the clinicians treating IBD patients. However, the study design is not clear enough to evaluate the meaning of the results. In addition, because of the small number in the stratified groups, the results require extreme caution to interpret. Furthermore, the presentation of the data is redundant and inefficient.
Major points:
1. The demographic data should be presented as a table. In addition, the first-presented numbers in the discussion are not acceptable.
2. Was the vaccination performed during the biologics treatment? The description like “exposed” imply that the group includes patients who were treated by biologics prior to the vaccination. If the patients were on the treatment at any of the three vaccine shots, that should be clarified. Otherwise, the study design is not clear.
3. The stratified groups other than anti-TNF therapy such as anti-integrin therapy have too small numbers of patients, which may not have enough statistical power to detect differences. That should be clearly stated.
4. The statistical method that was used for the multiple comparisons is not described. Is it Dunn’s multiple comparison? Were all the pairs evaluated? Or, the comparison between control group and each treatment was performed? I believe that the latter strategy is more suitable to have better power.
Minor points:
1. The title describes a speculation and thus sounds very weak. A title such as “Inefficient Induction of Neutralizing Antibodies against SARS-CoV-2 Variants in Patients with Inflammatory Bowel Disease on anti-Tumor Necrosis Factor Therapy Even after Receiving a Third mRNA Vaccine Dose” would sound more solid.
2. Table 1 and Figure 1a is largely redundant.
3. Table 2 and Figure 1b is largely redundant.
4. The wild type is located at the right most in Figure 1a, while it is located at the left most in Figure 1b. This is confusing.
4. In Figure 1b, the meaning of “IMM” is not described.
5. In line 123, the total number of anti-TNF therapy (17) is not equal to the sum of the adalimumab (10) and infliximab (8). Explain.
5. In line 134, Figure 2 should be Figure 1b.
6. In line 154, “demonstrate that ... might not ...” sounds inappropriate. Something demonstrated should be solid. I would recommend using “indicate”, instead.
Reviewer 3 Report
The article is very interesting since it explains the need for special observation of IBD patients under immunotherapy who are booster dosed for Covid 19. A few points I would like to cover that would help understand this study clearly.
1. The title of the paper needs to format since the the subjects did not have actual infection post third dose of mRNA vaccine.
2. In the materials and methods, ELISA and Viral Neutralization Assay should be separate titles.
3. In the results section, it would be great if the author could explain why 60-days has been chosen for blood sample collection.
4. The author also need to include a separate table for age, sex and BMI to conclude the study.
5. Not clearly written what was the average age of male and average age of female subjects.
6. Why there are only 12 patients in the control group?
7. What was the average age for the anti-TNF IBD patients who responded to third dose?
Round 2
Reviewer 2 Report
The raised concerns are solved.